# Prognostic Ability of Tumor Budding Outperforms Poorly Differentiated Clusters in Gastric Cancer

**DOI:** 10.3390/cancers14194731

**Published:** 2022-09-28

**Authors:** Luca Szalai, Ákos Jakab, Ildikó Kocsmár, Ildikó Szirtes, István Kenessey, Attila Szijártó, Zsuzsa Schaff, András Kiss, Gábor Lotz, Éva Kocsmár

**Affiliations:** 1Department of Pathology, Forensic and Insurance Medicine, Semmelweis University, Üllői Str. 93, H-1091 Budapest, Hungary; 2Department of Pathology, National Institute of Oncology, Ráth György Str. 7-9, H-1122 Budapest, Hungary; 3Department of Urology, Semmelweis University, Üllői Str. 78b, H-1082 Budapest, Hungary; 4National Cancer Registry, National Institute of Oncology, Ráth György Str. 7-9, H-1122 Budapest, Hungary; 5Department of Surgery, Transplantation and Gastroenterology, Semmelweis University, Üllői Str. 78, H-1091 Budapest, Hungary

**Keywords:** gastric cancer, tumor budding, poorly differentiated cluster, lymph node metastasis, survival

## Abstract

**Simple Summary:**

Small clusters of tumor cells at the invasion front of the tumor, known as “tumor budding” and larger clusters of cells, known as “poorly differentiated clusters”, are well known histological parameters to assess the outcome of colorectal cancer. However, their prognostic value for gastric cancer is less well investigated and controversial based on the studies conducted so far. In our retrospective study, we investigated the prognostic power of these two entities in a large cohort of gastric cancer patients. Our results demonstrate that tumor budding is an independent prognostic factor determining overall survival in gastric cancer, especially in intestinal type adenocarcinomas. In addition, budding can also predict the risk of lymph node metastases. However, although the trend is similar for poorly differentiated clusters, their predictive value is lower and they have not been shown to be an independent prognostic factor of survival in gastric cancer.

**Abstract:**

The prognostic value of histological phenomena tumor budding (TB) and poorly differentiated clusters (PDCs) have been less studied in gastric cancer (GAC) and the data provided so far are controversial. In our study, 290 surgically resected GAC cases were evaluated for TB according to the criteria of International Tumor Budding Consensus Conference (ITBCC) and PDC, and both parameters were scored on a three-grade scale as described for colorectal cancer previously (0: Grade0, 1–4: Grade1, 5–9: Grade2 and ≥10: Grade3) and classified as low (Grade0–2) and high (Grade3) TB/PDC. High TB/PDC was associated with diffuse-type morphology, higher pT status, incomplete surgical resection, poor tumor differentiation and perineural and lymphovascular invasion. Multivariable survival analyses have shown an independent prognostic role of high TB with poorer overall survival in the total cohort (*p* = 0.014) and in intestinal-type adenocarcinomas (*p* = 0.005). Multivariable model revealed high TB as an independent predictor for lymph node metastasis in both the total cohort (*p* = 0.019) and in the intestinal type adenocarcinomas (*p* = 0.038). In contrast to tumor budding, no significant association was found between PDC and the occurrence of lymph node metastasis and tumor stage and even survival. In conclusion, tumor budding is an independent prognostic factor of survival in gastric cancer, especially in intestinal-type adenocarcinomas.

## 1. Introduction

Although its incidence has been decreasing over the past half century, gastric cancer remains the fifth most common malignancy worldwide. Despite multimodal treatment options, it ranks fourth in the list of cancer deaths [1]. This is mainly because the tumor is often diagnosed late, at an advanced stage, due to the non-specific symptoms [2].

In gastric cancer, several conventional clinicopathological factors are known to be of prognostic and/or predictive significance. The most important prognostic factor is considered to be cancer stage as determined by the tumor/node/metastasis (TNM) system [3,4]. Other important prognostic factors include cancer grade (indicating the level of tumor differentiation, based mainly on glandular formation tendency in gastric cancer) [3,5], tumor size [3,5], the presence of lymphovascular invasion [3] and the intragastric localization of the tumor (proximal/gastroesophageal junction—GEJ/vs. distal type) [3]. Positive lymph node ratio (LNR) and neutrophil-lymphocyte ratio (NLR) are more recently accepted and less widely used histopathological prognostic factors, the higher values of both are associated with worse survival [6,7]. Additional factors, such as sex, age [3,5] and response to neoadjuvant chemotherapy are also of prognostic importance [8].

Due to its prognostic significance, the determination of tumor budding in colorectal cancer has become part of routine histopathological diagnostics, and therefore its prognostic role in gastric cancer has also been investigated by various research groups [9,10,11,12,13,14]. The term “tumor-budding” was first introduced by Imai and colleagues decades ago [15]. In its current definition, a tumor bud is defined as a single tumor cell or a cluster of up to four cohesive, non-lumen-forming tumor cells separated from the main tumor mass [16]. Their appearance, known as tumor budding, has been identified as an important prognostic histological feature of cancer. Elevated tumor budding activity directly correlates with tumor aggressiveness (poor differentiation and higher tumor stage) and the reduced overall survival of patients. Its prognostic role is based on the fact that tumor budding is a specific histological manifestation of epithelial-mesenchymal transition (EMT), a known hallmark of tissue invasiveness and thus ultimately metastasis, which is the major determinant of cancer patients’ life expectancy [17].

Although, as a result of international harmonization efforts, the International Tumor Budding Consensus Conference (ITBCC) has agreed that the upper limit for tumor bud size is four cells, this is an arbitrary cut-off [16]. Larger distinct solid tumor cell clusters are known as “poorly differentiated clusters (PDCs)” and their prognostic role, similar to that of tumor buds, was first investigated and demonstrated in colorectal cancer [18,19,20,21]. However, it is important to emphasize that these two entities (tumor bud and PDC) are indeed part of the same biological phenomenon. The number of distinct tumor cells/clusters appearing in the invasive front of the tumor is a quantitative, while their size is a qualitative characteristic of invasiveness and, therefore, their integration into a single scoring system has been proposed for squamous cell carcinomas [22,23,24]. In our previous study, in periampullary carcinomas (ampulla Vateri, duodenum, pancreatic head and distal bile duct adenocarcinomas), the presence of smaller PDCs (6–10 tumor cells) was identified as a negative prognostic factor, whereas the presence of larger tumor cell clusters (31–35 cells in size) was found to have a positive prognostic impact [25].

Moreover, compared to tumor budding, prognostic estimation based on PDC counting has several advantages. The accurate determination of budding may be problematic in certain cases because individual cancer cells and smaller tumor buds are sometimes difficult to recognize without cytokeratin immunohistochemical staining. In contrast, PDCs are more identifiable owing to their larger size, which may be particularly useful in certain cases, such as when tumor cells/clusters need to be counted adjacent to degenerating normal glands or in the presence of a dense inflammatory background [20]. In colorectal carcinomas, moreover, the presence of PDCs has been shown by different studies to be a stronger prognostic predictor than previously accepted tumor budding [26,27,28].

Since only a limited number of studies have been published on this topic in gastric cancer, we have aimed to investigate the prognostic significance of tumor budding assessed according to the ITCCB criteria in a single institutional gastric cancer cohort, and also to evaluate the prognostic power of PDC counting in the same cohort, as established by Ueno and colleagues in colorectal cancer [27]. Accordingly, the overall survival (OS) was the primary endpoint of our study for both tumor bud and PDC, while disease free survival (DFS) as well as the occurrence of lymph node or distant metastases were secondary endpoints. In the following, we show that tumor budding has been found to be an independent prognostic factor for overall survival in gastric cancer, whereas PDC, although showing a similar trend, has had a lower prognostic significance.

## 2. Methods

### 2.1. Study Population and Case Selection

In this retrospective study, all gastric adenocarcinoma patients treated with primary total or partial gastrectomy between 2008 and 2018 were selected from the institutional archive of the Department of Pathology, Forensic and Insurance Medicine, Semmelweis University, Budapest, Hungary. Of these, diagnostic glass slides were available for analysis in 374 cases of gastric adenocarcinoma. The following exclusion criteria were applied: (1) patients whose data were incomplete, (2) patients who underwent previous chemotherapy before surgery and (3) tumor budding was not assessable due to microenvironmental factors that inhibited the detection of individual dissociated tumor cells or small tumor cell clusters (see in Appendix A). Finally, a total of 290 patients were included in the study. From each patient, multiple hematoxylin and eosin (HE) stained sections were examined by light microscopy and a representative section with the highest density of tumor buds was used for further analysis. Selected histological slides were scanned and digitized (Panoramic 1000 Digital Slide Scanner, 3D Histech, Budapest, Hungary), and tumor budding and PDCs were assessed on digital slides by visualizing them with CaseViewer software (3D Histech, Budapest, Hungary) independently by two specially trained researchers (L.S. and Á.J.) blinded to clinical and outcome data. In case of substantially discrepant results, the given case was jointly assessed again in a second round involving further experienced investigator(s) (É.K. and G.L.). The detailed clinical anamnesis of the patients were collected from the electronic patient register of Semmelweis University. The dates and causes of deaths were obtained from the Hungarian Cancer Registry, National Institute of Oncology.

### 2.2. Ethical Permission

The study protocol was following the ethical guidelines of the 1975 Declaration of Helsinki and was approved by the Ethical Committee of the Semmelweis University, Budapest (SE-RKEB 245/2019). Based on the current Hungarian law for scientific research, contacting the patients in order to have their informed consent is basically not requested for the retrospective studies. According to this, the Ethical Committee of the Semmelweis University, Budapest has waived the informed consent procedure for the study.

### 2.3. Histological Assessment of Tumor Budding and Poorly Differentiated Clusters

The assessment of the TBs and PDCs were performed in HE-stained digitalized slides according to the recommendations of ITBCC [16]. Accordingly, the TBs and PDCs were evaluated in the invasive front of each tumor using the hot-spot method, which is considered to be the most useful approach for assessing tumor budding in colorectal cancer. TB and PDC count was assessed in the standardized field area of 0.785 mm^2^ at 200× total magnification. The resulting number of TBs and PDCs per 0.785 mm^2^ high power field is used for further analysis. According to the recommendation of ITBCC, tumor buds were specified as a single individual tumor cell or as a solid cluster of up to four tumor cells and cases were classified into budding grades based on the total number of buds in the investigated hot spot area (Bd 0: 0, Bd 1: 1–4, Bd 2: 5–9, Bd 3: ≥10 TBs, respectively). We also assessed the poorly differentiated clusters (PDCs), which were defined as a solid tumor cell cluster of five or more cells [27]. PDC grades were determined similarly to the TB grades based on the total number of PDCs in the investigated hot spot area according the recommended scoring system by Ueno and his colleagues and with the addition of a new class for cases with zero PDC (PDC 0: 0, PDC 1: 1–4, PDC 2: 5–9, PDC 3: ≥10 PDCs, respectively) [27]. The adenocarcinoma subtypes were defined according to the Lauren classification [29] and the pTNM stage was determined following the 8th edition of the AJCC guidelines [30].

### 2.4. Statistical Analysis

All calculations and plotting were carried out in R software environment (version 4.0.2) and R Studio portable (version 1.3.959). As in previous studies, the Bd0, Bd1 and Bd2 groups were combined in the calculations to “TB low”, while the Bd3 group alone represented the “TB high” category. PDC groups were classified into “DC low” and “PDC high” categories in the same way. In order to ensure greater statistical power in the calculation of hazard/odds ratios in multivariable analysis, not all possible increments were included in the model in the case of pT and pN but similarly to other studies [4,6], to also take into account clinical relevance, pT stages I–II and III–IV were combined, so that a two-stage scale (I–II vs. III–IV) was used instead of four, and a similar two-stage (pN0 vs. pN+) scale was used for pN. During the statistical analysis, continuous variables were described as means, range and SD and categorical variables as frequencies. Categorical data were analyzed using 2 × 2 or 2 × 3 contingency tables and compared using Fisher’s exact probability test. Survival analyses were performed using the Kaplan–Meier/log-rank method, as well as the correlation between OS or DFS and TB or PDC scores were analyzed using univariable and multivariable Cox proportional hazards regression with backward stepwise selection. Patients who died within 30 days of surgery were not included in the survival analyses (perioperative death; *n* = 16/290; 5.52%). Univariable and multivariable logistic regression analyses with backward selection method were used to identify the independent risk factors for occurrence of lymph node metastasis. The correlations between LNR and Bd or PDC groups were tested pairwise using Wilcoxon’s test, and Kruskal–Wallis test was used to compare all groups in one step. In multiple comparisons, *p* values were adjusted by applying the Benjamini–Hochberg method. All *p* values were calculated two-tailed and considered significant when *p* < 0.05.

## 3. Results

### 3.1. Cohort Characteristics

A total of 290 patients were selected for the study, with a gender distribution of 107 females and 183 males. The clinicopathological characteristics of the cohort are shown in Table 1. The mean age was 66.13 years (range 34–89). In all the surgery performed, 184 (63.45%) cases were total gastrectomy and 106 (36.55%) patients underwent subtotal gastric resection. The tumor was located in the cardia in 98 cases (33.79%) and in the distal region of the stomach in 192 (66.21%) cases. According to Lauren’s classification of gastric cancers, a total of 159 (54.83%) cases of intestinal morphology, 107 (36.90%) cases of diffuse morphology and 27 (9.31%) cases of mixed morphology were identified. A total of 222 cases showed tumor-free resection margins (R0; 76.55%) during histological examination, while 68 cases underwent incomplete resection (R1; 23.45%). According to the 8th TNM classification [30], 8.97% (26 cases) of the patients were classified as pT1, 7.24% (21 cases) as pT2, 50.35% (146 cases) as pT3 and 33.45% (97 cases) were at pT4 stages. A total of six cases (2.07%) were graded as well differentiated (G1), 92 (31.72%) tumors as moderate differentiated (G2) and 192 (66.21%) tumors as undifferentiated (G3). Lymphovascular invasion was present in 103 cases (35.52%) while perineural invasion was observed only in 81 cases (27.93%). Median follow-up was 34 months (ranged 1–137 months). The patients were divided into TB groups according to the ITBCC guideline with the slight modifications of Zlobec and her collaegues [31], which resulted in three patients (1.03%) being classified in Bd 0, 19 (6.55%) patients in Bd 1, 40 (13.79%) patients in Bd 2 and 228 (78.62%) patients in Bd 3 (Figure 1). According to the classification of patients by PDC grade, a total of one (0.35%) patient was classified as PDC 0, 16 (5.52%) patients as PDC 1, 39 (13.45%) patients as PDC 2 and 234 (80.69%) patients as PDC 3 (Figure 1).

### 3.2. Association of Tumor Budding and PDCs with Clinicopathological Parameters

The associations between clinicopathological parameters and TB or PDC low/high status in the cohort are shown in Table 2. Tumor budding was correlated with Lauren-phenotype, incomplete surgical resection, presence of perineural and lymphovascular invasion, pT status, grade, UICC stage, the presence of lymph nodes and distant metastasis. High PDC category showed a significant correlation with the Lauren-phenotype, residual tumor after surgery, the presence of perineural and lymphovascular invasion and pT status and grade.

### 3.3. Survival Analysis

The results of univariable and multivariable survival analyses in the total cohort and in intestinal type adenocarcinomas are shown in Table 3 and Table 4, respectively. In the total cohort, median overall survival (OS) was 17 months and median disease free survival (DFS) was 13 months. In univariable analysis, the overall survival of the patients significantly depended on TB low/high category, sex, pT/pN/pM status, the presence of residual tumor and lymphovascular or perineural invasion. In multivariable Cox regression model, TB category, sex, age, pN status and incomplete surgical resection were found as independent negative prognostic factors of overall survival. In univariable setting, reduced disease free survival was significantly associated with TB high category, sex, pT high/pN+/pM1 status and the presence of residual tumor, lymphovascular or perineural invasion. However, upon multivariable analysis, the TB high category was not found to be an independent prognostic factor for disease free survival.

The median survival of patients suffering from intestinal type adenocarcinoma was 24 months, the disease free survival was 16 months. In intestinal type adenocarcinomas overall survival was significantly associated with TB category, sex, pT, pN and pM status, as well as the presence of residual tumor, perineural and lymphovascular invasion. In the multivariable model, TB high category, pN+ status and incomplete surgical resection were independently associated as negative prognostic factors for overall survival. Univariable analysis indicated that overall survival in the intestinal cohort was significantly related to TB category, age, pT/pN/pM status, presence of residual tumor, lymphatic and perineural invasion. In multivariable analysis, however, high TB category was not identified as an independent prognostic factor for disease free survival.

For diffuse type adenocarcinomas, neither TB nor PDC was found to be an independent prognostic factor for survival (see in Appendix A).

Kaplan–Meier survival curves and log-rank analysis also indicated a significantly worse OS and DFS for high budding cases in both the total cohort and intestinal type adenocarcinomas, while no significant difference was found between the PDC low and PDC high categories (Figure 2). No significant difference was observed between TB low and TB high, as well as between PDC low and PDC high categories in diffuse type adenocarcinomas (see in Appendix A).

### 3.4. Association of Lymph Node Metastasis with Tumor Budding and Poorly Differentiated Clusters

The occurrence of lymph node metastases was analyzed among patients with low and high TB as well as low and high PDC categories, respectively. Univariable logistic regression showed a significant positive correlation between pN positive status and the low/high TB category, pT and pM stages, Lauren subtype, grade, positive resection margin and the presence of lymphovascular or perineural invasion (Table 5). Multivariable logistic regression analysis with backward selection revealed high tumor budding as an independent predictor for lymph node metastasis in both the total cohort (*p* = 0.019, OR: 2.86, CI: 1.197–6.877) and in the intestinal type adenocarcinomas (*p* = 0.038, OR: 2.778, CI: 2.78–7.52). The trend was similar, but no significant association was found between high PDC and the presence of lymph node metastasis (total cohort: *p* = 0.057; intestinal cohort: *p* = 0.075) (Table 5).

Furthermore, the tumorous involvement of lymph nodes in patients of different Bd and PDC grades was also investigated. The ratio of positive lymph nodes to total lymph nodes excised (LNR) was significantly correlated with 0 + 1/2/3 Bd grades in both the total cohort (*p* < 0.0001) as well as in intestinal and diffuse type cancers (*p* = 0.012 in intestinal cancers and *p* = 0.03 in diffuse cancers, respectively). Grading for PDC showed a significant correlation with LNR only in the total cohort (*p* = 0.02) and in diffuse type cancers (*p* = 0.011) (Figure 3).

## 4. Discussion

The results of our study confirm that tumor budding, assessed by the number of individual tumor cells or small (2–4 cell) solid tumor cell nests separated from the main tumor mass according to the ITBCC criteria, predicts the development of lymphatic metastasis in gastric cancer and is an independent prognostic factor for overall survival, especially in intestinal type gastric cancer. On the other hand, the occurrence of larger tumor cell clusters (PDCs) has shown an association with the features of local tumor spread but is not predictive of survival or lymph node metastasis.

As the ITBCC guideline specifies a tumor bud size up to four tumor cells, solid tumor cell nests larger than this are defined as a poorly differentiated cluster (PDC). However, these two entities (tumor bud and PDC) are in fact part of the same biological spectrum and therefore both may provide similar prognostic data. This is supported by the fact that these tumor cells, which appear to be detached from the main tumor mass, usually represent distinct tumor cell clusters only in the two-dimensional plane of histological sections but most of them have been found to be connected to the main tumor mass by three-dimensional spatial reconstruction [32]. In fact, for colorectal tumors, the prognostic power of PDCs has been demonstrated to be even stronger than that of tumor buds, and for squamous carcinomas, the integration of tumor budding and PDC into a single scoring system has been proposed. One of our previous studies showed an opposite prognostic predictive role for PDCs of different sizes in periampullary carcinomas, with smaller PDCs being identified as a negative while the presence of larger tumor cell clusters as a positive prognostic factor [25].

Molecular markers and molecular subtype classifications have opened up new possibilities in the prognostic and predictive stratification of malignant tumors. However, in contrast to, e.g., breast cancer, their full implementation in the routine diagnostic practice of gastric cancer has not yet been achieved, because their prognostic/predictive significance is not explicit for all subtypes. A further factor is that gastric cancer subtyping is not fully amenable using immunohistochemical tests in routine pathology practice, with the exception of microsatellite instability testing by MMR immunohistochemistry, which has a strong predictive value for immune checkpoint inhibitor (ICI) treatments [33]. Subgroup determination by molecular methods is still costly and requires additional testing time. A major advantage of the prognostic stratification based on tumor bud and/or PDC counting is that it can be determined on routine HE-stained histological sections simultaneously with the primary tumor diagnosis without extra costs. Further treatment or follow-up can then be planned, taking this into account.

Tumor budding is considered to be a special form of epithelial-mesenchymal transition (EMT) [17]. Previous studies have shown that EMT is associated with the activation of various immune checkpoint molecules, including PD-L1. Recently, a positive correlation between the EMT markers of mesenchymal transition and tumor PD-L1 expression has also been demonstrated [34]. This EMT-induced mechanism of tumor cell escape from immune surveillance promotes cancer progression and may therefore be a potential predictor of response to immune checkpoint inhibitor (ICI) therapy. This may be particularly significant because PD-L1 expression alone has not been established as a predictive marker of ICI therapy in gastric cancer. Although the FDA previously approved pembrolizumab treatment for gastric adenocarcinomas with CPS ≥ 1 PD-L1 expression, its use is no longer linked to PD-L1 positivity. However, EMT is definitely a phenotypic alteration required for tissue invasiveness and the ability to metastasize, which may explain why a positive correlation between tumor budding and the appearance of lymph node metastases has been found.

Only a limited (but fortunately growing) number of studies have been conducted on the prognostic significance of tumor budding and PDCs in gastric cancer. Although previous studies have already demonstrated a strong correlation between tumor budding and poorer survival as well as the occurrence of lymph node metastases [10,12,14,35], the introduction of the ITBCC guideline for colorectal cancer has provided a standard method of tumor budding assessment that has given a new stimulus to further study this phenomenon also in gastric cancer. In their previous study, Gabbert and colleagues used a semi-quantitative approach to classify their cases into grades one to four based on the degree of “tumor cell dissociation”, used synonymously to describe tumor budding. Similarly to our present study, their analysis proved the higher density of single tumor cells detached from the main tumor mass to be an independent prognostic factor for survival in intestinal-type adenocarcinomas [14]. Moreover, in recent years, further important studies have been published [9,11,13], in which budding assessed according to the ITBCC criteria has been shown to be prognostic for survival in gastric cancer, which is particularly true for intestinal type gastric cancer. Interestingly, only the studies by Kemi et al. [9] and Dao et al. [11] were able to identify budding as an independent prognostic factor, but Ulase et al. [13] were unable to demonstrate its independence from other prognostic variables in multivariable analyses. The same study demonstrated that not only the occurrence of lymph node metastasis itself, but also the lymph node ratio (LNR, i.e., the proportion of regional lymph nodes that are metastatic) was significantly associated with tumor budding [13]. Our own results fully support these findings, as in addition to demonstrating tumor budding as an independent prognostic factor for survival in the whole cohort and in the intestinal-type adenocarcinoma, high budding was significantly associated not only with local tumor spread features (pT and stage, lymphovascular and perineural invasion) but also with the occurrence of lymph node metastases and LNR.

It is an interesting observation that, in addition to lymph node status and residual tumor, tumor budding is an independent prognostic factor for overall survival in both the total cohort and intestinal adenocarcinomas. However, in contrast to OS, DFS showed a significant association with budding only in univariable analysis; however, in multivariable analysis, besides lymph node status and residual tumor, only distant metastasis (the latter only in the total cohort) was found to be an independent prognostic factor for DFS. Besides tumor budding, pT (extent of primary tumor) and lymphovascular invasion have been identified as independent prognostic factors for lymph node metastasis. The possibility of predicting lymph node involvement by tumor budding is also significant because in the case of endoscopic mucosectomy and endoscopic submucosal dissection, the budding of the primary tumor can be used to select cases requiring conventional (partial or total) gastrectomy, as the probability of lymph node metastases is high. To the best of our knowledge, the prognostic significance of PDC in gastric cancer has only been investigated in one study with a small number of cases (*n* = 50) [36]. They observed a significant correlation between PDC categories and lymphovascular invasion, lymph node metastasis, LNR and stage, as well as a significant association with poor disease free and overall survival. In our study, however, in contrast to tumor budding, we found no significant correlation between PDC and the occurrence of lymph node metastasis and (probably therefore) tumor stage as well as, more importantly, survival. With several other variables that showed statistically significant associations with tumor budding (Lauren classification (intestinal/diffuse tissue pattern), residual tumor infiltration of the resection margin, perineural invasion, lymphovascular invasion, pT (invasion depth), tumor differentiation level) PDC was also correlated in our study, although it was close to the limits of significance for resection margin, perineural invasion and lymphovascular invasion. This both supports the theory that PDCs have similar prognostic capabilities to tumor buds and emphasizes that tumor buds are a better predictor of prognosis than PDCs in gastric cancer. It is important to note that PDCs displayed only marginally significant correlation with just those features, such as perineural and lymphovascular invasion, which are determinants of subsequent recurrence and metastatic spread, respectively [37,38,39]. In a previous study, Tanaka and colleagues found that perineural invasion is a strong prognostic factor for five-year survival in gastric cancer. In our study, the presence of perineural invasion was significantly associated with both overall and disease-free survival in univariate analysis. However, in contrast to tumor budding, multivariate analysis did not reveal the perineural invasion to be an independent prognostic factor of survival in either the entire cohort or only in intestinal-type adenocarcinomas [37]. This is likely to be reflected in the fact that PDCs have shown no significant association with either lymph node metastases (and hence stage) or overall survival. Consistent with these findings, Jun et al. found a significant association of intratumoral/peritumoral tumor budding and PDCs with survival for all these four EMT features in small bowel adenocarcinomas but similar to the results of our present study, only peritumoral tumor budding was found to be an independent prognostic factor for survival in multivariate analysis [40].

As with retrospective, single-center studies in general, the present study has limitations. As there were few cases of pT1 in our cohort, we were unable to specifically investigate the ability of tumor budding to predict the development of lymph node metastasis in early gastric cancer. However, this issue has been studied in detail by Gulluoglu et al. [10] and Ulase et al. [13] who also found a significant association between the occurrence of lymph node metastasis and high tumor budding in T1 cancers. These findings suggest that in the case of the high tumor budding of the primary tumor in endoscopic mucosectomy or endoscopic submucosal dissection specimen, a subsequent gastrectomy may be worth considering. A further limitation of our study is that an accurate determination of DFS was not always possible because information on the exact status of some patients who were still alive was not available in the existing databases.

## 5. Conclusions

In conclusion, our results demonstrate that in addition to lymph node status and residual tumor, tumor budding assessed according to ITBCC criteria is also an independent prognostic factor determining survival in gastric cancer, especially in intestinal type adenocarcinomas. In addition, budding can also predict the likelihood of lymph node metastases. However, although the trend is similar for PDCs, their predictive ability is lower, and they have not been shown to be an independent prognostic factor in gastric cancer. Based on our results, the introduction of prognostic stratification according to the ITBCC recommendations should be considered for gastric adenocarcinomas in addition to colorectal tumors.

## Figures and Tables

**Figure 1 cancers-14-04731-f001:**
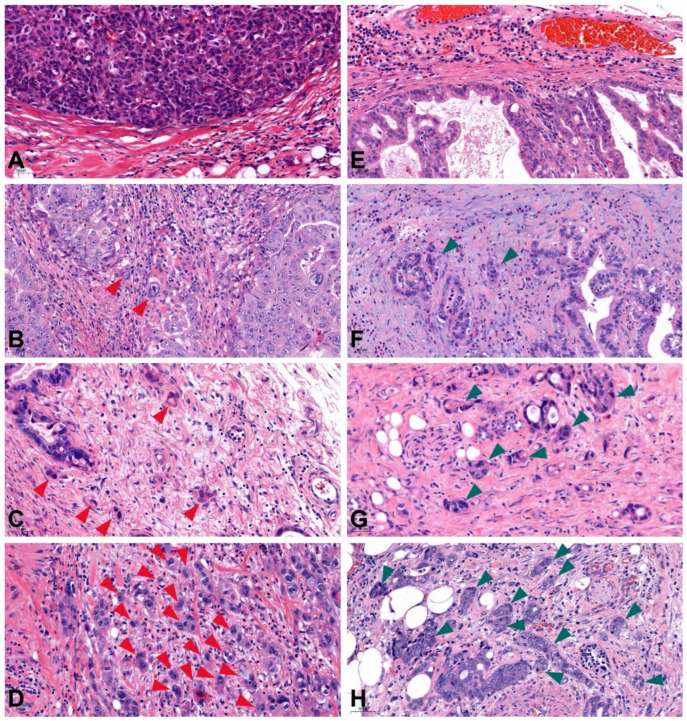
Tumor budding and PDC grades assessed according to ITBCC recommendations. Tumor buds are indicated by red arrowheads, PDCs by green arrowheads. (**A**) Bd grade 0 (no tumor bud in the hot spot), (**B**) Bd grade 1 (1–4 tumor bud/hot spot), (**C**) Bd grade 2 (5–9 tumor bud/hot spot), (**D**) Bd grade 3 (≥10 tumor bud/hot spot), (**E**) PDC grade 0 (no PDC in the hot spot), (**F**) PDC grade 1 (1–4 PDC/hot spot), (**G**) PDC grade 2 (5–9 PDC/hot spot), (**H**) PDC grade 3 (≥10 PDC/hot spot); (PDC: poorly differentiated cluster, original magnification 20×).

**Figure 2 cancers-14-04731-f002:**
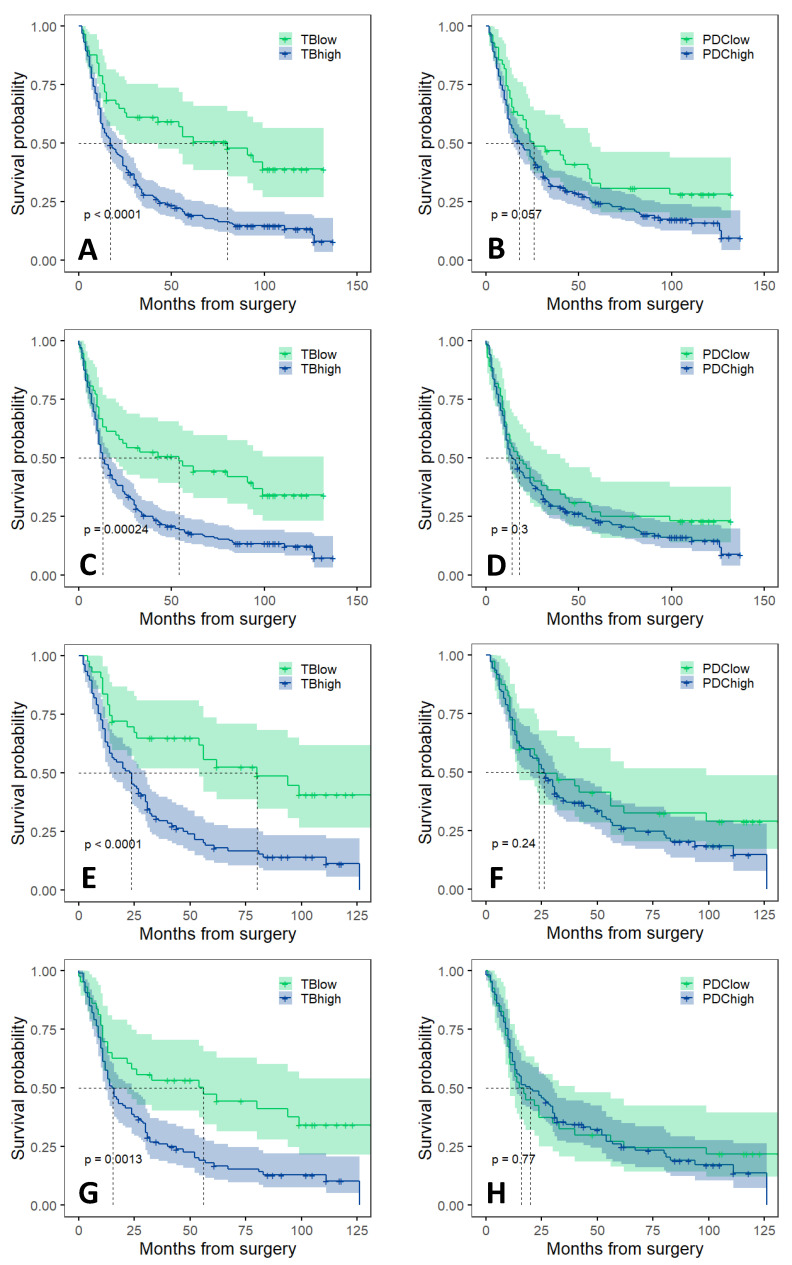
Kaplan–Meier curves of OS and DFS survival analyses. (**A**) OS by TB low and TB high groups in the total cohort, (**B**) OS by PDC low and PDC high groups in the total cohort, (**C**) DFS by TB low and TB high groups in the total cohort, (**D**) DFS by PDC low and PDC high groups in the total cohort, (**E**) OS by TB low and TB high groups in the intestinal type adenocarcinomas, (**F**) OS by PDC low and PDC high groups in the intestinal type adenocarcinomas, (**G**) DFS by TB low and TB high groups in the intestinal type adenocarcinomas, (**H**) DFS by PDC low and PDC high groups in the intestinal type adenocarcinomas. OS: overall survival, DFS: disease free survival, TB: tumor budding, PDC: poorly differentiated cluster.

**Figure 3 cancers-14-04731-f003:**
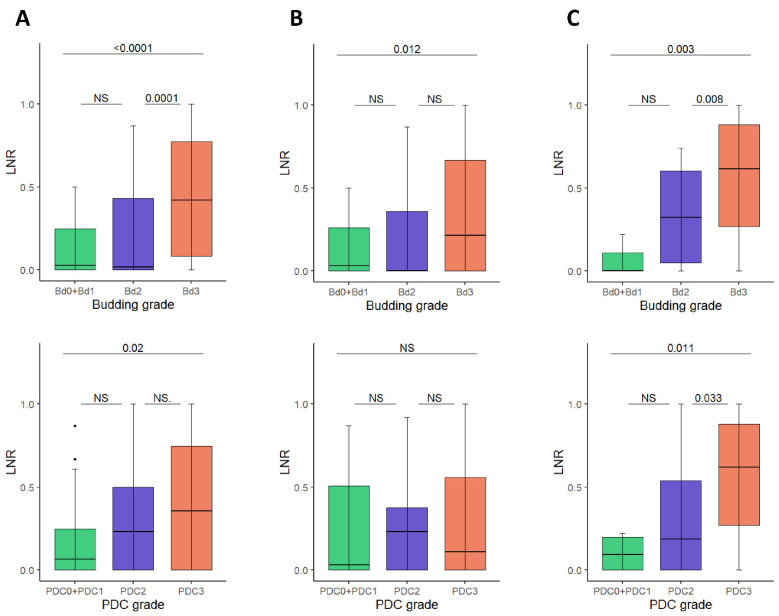
Association between the lymph node ratio (LNR) and tumor budding/PDC grades. (**A**) Total cohort, (**B**) Intestinal type gastric cancers, (**C**) Diffuse type gastric cancers. PDC: poorly differentiated clusters.

**Table 1 cancers-14-04731-t001:** Cohort characteristics. (pT: Primary tumor extent, pN: Regional lymph node metastasis, LNR: lymph node ratio, UICC: Union for International Cancer Control, PDC: poorly differentiated cluster, pM: Distant metastasis).

Variable		*n*/%
Number of patients	*n*	290
Age at surgery (years)	mean, range	66.12759 (34–89)
Sex	female	107
	male	183
Extent of gastrectomy	total	184
	subtotal	106
Location	cardia	98
	fundus–body–antrum	192
Lauren classification	intestinal	159
	diffuse	107
	mixed	27
Resection margin	R0	222
	R1	68
Perineural invasion	present	81
	absent	209
Lymphovascular invasion	present	179
	absent	111
Tumor differentiation	well	6
	moderate	92
	poor	192
pT	Ia	12
	Ib	14
	II	21
	III	146
	IVa	73
	IVb	24
pN	0	80
	1	52
	2	56
	3a	62
	3b	40
LNR	mean, range	0.38 (0.0–1.0)
Distant metastasis (pM)	present	17
	absent	273
UICC Stage	IA	23
	IB	15
	IIA	35
	IIB	49
	IIIA	52
	IIIB	57
	IIIC	42
	IV	17
Tumor budding grade	Bd 0	3
	Bd 1	19
	Bd 2	40
	Bd 3	228
PDC grade	PDC 0	1
	PDC 1	16
	PDC 2	39
	PDC 3	234
Follow-up period	mean, range	33.803 (1–137)
Death	*n*	228 (78.62%)
Perioperative death	*n*	16/290 (5.52%)

**Table 2 cancers-14-04731-t002:** Association between clinicopathological parameters and TB or PDC low/high status in gastric adenocarcinoma (pT: Primary tumor extent, pN: Regional lymph node metastasis, pM: Distant metastasis, UICC: Union for International Cancer Control, n.s.: statistically non-significant; Statistically significant *p* values are displayed in bold).

Parameter		Tumor Budding			Poorly Differentiated Cluster		
		Low (0 + 1 + 2)	High (3)	*p*	Low (0 + 1 + 2)	High (3)	*p*
Number of patients (*n*)	*n*	62 (21.38%)	228 (78.62%)	-	56	234	-
Age (mean, range)	mean (years), range	68.4 (49–87)	65.5 (34–89)	n.s.	66.446	66.051	n.s.
Sex	female	23 (37.1%)	84 (36.8%)	1	18 (32.1%)	89 (38.0%)	n.s.
	male	39 (62.9%)	144 (63.2%)		38 (67.9%)	145 (62.0%)	
Extent of gastrectomy	total	34 (54.8%)	150 (65.8%)	n.s.	35 (62.5%)	149 (63.7%)	n.s.
	subtotal	28 (45.2%)	78 (34.2%)		21 (37.5%)	85 (36.3%)	
Location	cardia	17 (27.4%)	81 (35.5%)	n.s.	19 (33.9%)	79 (33.8%)	n.s.
	distal	45 (75.6%)	147 (64.5%)		37 (66.1%)	155 (66.2%)	
Lauren classification	intestinal	48 (77.4%)	111 (48.7%)	**0.0003**	41 (73.2%)	118 (50.4%)	**0.0086**
	diffuse	11 (17.7%)	96 (42.1%)		13 (23.2%)	94 (40.2%)	
	mixed	3 (4.8%)	21 (9.2%)		2 (3.6%)	22 (9.4%)	
Resection margin	R0	55 (88.7%)	167 (73.2%)	**0.0174**	49 (87.5%)	173 (73.9%)	**0.0480**
	R1	7 (11.3%)	61 (26.8%)		7 (12.5%)	61 (26.1%)	
Perineural invasion	present	10 (16.1%)	71 (31.1%)	**0.0295**	9 (16.1%)	72 (30.8%)	**0.0417**
	absent	52 (83.9%)	157 (68.9%)		47 (83.9%)	162 (69.2%)	
Lymphovascular invasion	present	28 (45.2%)	151 (66.2%)	**0.0032**	28 (50.0%)	151 (64.5%)	**0.0481**
	absent	34 (54.8%)	77 (33.8%)		28 (50.0%)	83 (35.5%)	
pT	I–II	22 (35.5%)	25 (11.0%)	**<0.0001**	38 (67.9%)	29 (12.4%)	**<0.0001**
	III–IV	40 (64.5%)	203 (89.0%)		18 (32.1%)	205 (87.4%)	
Tumor differentiation	well	5 (8.1%)	1 (0.4%)	**<0.0001**	4 (7.1%)	2 (0.9%)	**<0.0001**
	moderate	30 (48.4%)	62 (27.2%)		30 (53.6%)	62 (26.5%)	
	poor	27 (43.5%)	165 (72.4%)		22 (39.3%)	170 (72.6%)	
UICC Stage	I–II	42 (67.7%)	80 (35.1%)	**<0.0001**	27 (48.2%)	95 (40.6%)	n.s.
	III–IV	20 (32.3%)	148 (64.9%)		29 (51.8%)	139 (59.4%)	
Lymph node metastasis (pN)	present	31 (50%)	179 (78.5%)	**<0.0001**	36 (64.3%)	174 (74.4%)	n.s.
	absent	31 (50%)	49 (21.5%)		20 (35.7%)	60 (25.6%)	
Distant metastasis (pM)	M0	61 (98.4%)	212 (93.0%)	n.s.	52 (92.9%)	221 (94.4%)	n.s.
	M1	1 (1.6%)	16 (7.0%)		4 (7.1%)	13 (5.6%)	

**Table 3 cancers-14-04731-t003:** Univariable and multivariable survival analyses in the total cohort (pT: Primary tumor extent, pN: Regional lymph node metastasis, pM: Distant metastasis, n.s.: statistically non-significant; Statistically significant *p* values are displayed in bold).

Parameter		Total Cohort—Overall Survival				Total Cohort—Disease Free Survival			
		Univariable		Multivariable		Univariable		Multivariable	
		HR (95% CI)	*p*	HR (95% CI)	*p*	HR (95% CI)	*p*	HR (95% CI)	*p*
TB	low/high	2.3 (1.5–3.3)	<0.0001	1.65 (1.11–2.45)	0.014	2 (1.4–2.8)	0.0003	n.s.	n.s.
PDC	low/high	1.4 (0.99–2)	0.058	n.s.	n.s.	1.2 (0.85–1.7)	0.3	n.s.	n.s.
Sex	M/F	0.72 (0.54–0.96)	0.025	0.72 (0.54–0.96)	**0.023**	0.8 (0.61–1.1)	0.12	n.s.	n.s.
Age	years	1 (0.99–1)	0.41	1.02 (1.00–1.03)	**0.009**	1 (0.99–1)	0.94	n.s.	n.s.
pT	I–II/ III–IV	2.6 (1.7–4)	<0.0001	n.s.	n.s.	2.7 (1.8–4.1)	<0.0001	n.s.	n.s.
pN	pN0/pN+	3 (2.1–4.2)	<0.0001	2.79 (1.96–3.96)	**<0.0001**	2.9 (2.1–4)	<0.0001	2.73 (1.96–3.79)	**<0.0001**
pM	pM0/pM1	2.1 (1.3–3.6)	0.036	n.s.	n.s.	2.6 (1.6–4.2)	0.0002	2.05 (1.24–3.39)	**0.005**
Lauren-type	intestinal/ other	1.1 (0.93–1.4)	0.23	n.s.	n.s.	1.1 (0.91–1.3)	0.33	n.s.	n.s.
Grade	G1/G2/G3	1.2 (0.93–1.6)	0.16	n.s.	n.s.	1.2 (0.89–1.5)	0.29	n.s.	n.s.
Residual tumor	absent/ present	2 (1.5–2.8)	<0.0001	1.84 (1.33–2.54)	**0.0002**	1.9 (1.4–2.6)	<0.0001	1.65 (1.21–2.27)	**0.002**
Lymphovascular invasion	absent/ present	2.1 (1.6–2.8)	<0.0001	n.s.	n.s.	2.1 (1.6–2.7)	<0.0001	n.s.	n.s.
Perineural invasion	absent/ present	1.5 (1.1–1.9)	0.011	n.s.	n.s.	1.5 (1.1–2)	0.0035	n.s.	n.s.

**Table 4 cancers-14-04731-t004:** Univariable and multivariable survival analyses in the intestinal type adenocarcinomas (pT: Primary tumor extent, pN: Regional lymph node metastasis, pM: Distant metastasis, n.s.: statistically non-significant; Statistically significant *p* values are displayed in bold).

Parameter		Intestinal Type—Overall Survival				Intestinal Type—Disease Free Survival			
		Univariable		Multivariable		Univariable		Multivariable	
		HR (95% CI)	*p*	HR (95% CI)	*p*	HR (95% CI)	*p*	HR (95% CI)	*p*
TB	low/high	2.5 (1.6–4.1)	0.0001	**1.99** **(1.23–3.22)**	**0.005**	2 (1.3–3.2)	0.0016	n.s.	n.s.
PDC	low/high	1.3 (0.84–2)	0.24	n.s.	n.s.	1.1 (0.7–1.6)	0.78	n.s.	n.s.
Sex	M/F	0.67 (0.44–1)	0.058	n.s.	n.s.	0.82 (0.55–1.2)	0.34	n.s.	n.s.
Age	years	0.99 (0.98–1)	0.59	n.s.	n.s.	0.98 (0.96–1)	0.07	n.s.	n.s.
pT	I–II/ III–IV	1.7 (1–2.8)	0.034	n.s.	n.s.	1.8 (1.1–2.9)	0.013	n.s.	n.s.
pN	pN0/pN+	2.6 (1.7–4)	<0.0001	**2.40** **(1.57–3.66)**	**<0.0001**	2.4 (1.6–3.6)	<0.0001	**2.46** **(1.65–3.66)**	**<0.0001**
pM	pM0/pM1	2.9 (1.3–6.2)	0.007	n.s.	n.s.	2.3 (1.1–5.1)	0.03	n.s.	n.s.
Grade	G1/G2/G3	1 (0.71–1.4)	0.96	n.s.	n.s.	0.96 (0.69–1.3)	0.8	n.s.	n.s.
Residual tumor	absent/ present	2.2 (1.4–3.4)	0.001	**2.04** **(1.28–3.26)**	**0.0028**	2 (1.2–3.1)	0.0035	**2.08** **(1.31–3.30)**	**0.0018**
Lymphovascular invasion	absent/ present	2.2 (1.5–3.3)	<0.0001	n.s.	n.s.	2.1 (1.5–3.1)	<0.0001	n.s.	n.s.
Perineural invasion	absent/ present	1.8 (1.2–2.8)	0.006	n.s.	n.s.	1.9 (1.2–2.8)	0.0031	n.s.	n.s.

**Table 5 cancers-14-04731-t005:** Lymph node metastasis prediction (pT: Primary tumor extent, pM: Distant metastasis, n.i.: not included in the analysis /as only intestinal type was analysed in these columns, Lauren type was not included as a variable here/, n.s.: statistically non-significant; Statistically significant *p* values obtained from multivariable analyses are displayed in bold).

Parameter		Total Cohort				Intestinal Type			
		Univariable		Multivariable		Univariable		Multivariable	
		OR (95% CI)	*p*	OR (95% CI)	*p*	OR (95% CI)	*p*	OR (95% CI)	*p*
TB	low/high	3.65 (2.03–6.62)	<0.0001	**2.86** **(1.20–6.88)**	**0.019**	2.36 (1.18–4.76)	0.015	**2.78** **(1.07–7.52)**	**0.038**
PDC	low/high	1.61 (0.86–2.98)	0.132	**0.37** **(0.13–0.99)**	**0.057**	0.94 (0.44–1.94)	0.860	n.s.	0.075
Sex	M/F	1.12 (0.66–1.93)	0.68	n.s.	n.s.	1.25 (0.62–2.60)	0.534	n.s.	n.s.
Age	years	0.98 (0.95–1.00)	0.051	n.s.	n.s.	0.96 (0.92–0.99)	0.009	**0.96** **(0.92–0.99)**	**0.036**
pT	I–II/ III–IV	14.80 (7.20–32.65)	<0.0001	**7.19** **(3.09–17.75)**	**<0.0001**	7.58 (3.24–19.49)	<0.0001	**4.34** **(1.65–12.32)**	**0.004**
pM	pM0/pM1	6.52 (1.30–118.59)	0.071	n.s.	n.s.	3.81 (0.63–72.85)	0.221	n.s.	n.s.
Lauren-type	intestinal/ other	2.23 (1.42–3.63)	0.0008	n.s.	n.s.	n.i.	-	n.i.	-
Grade	G1/G2/G3	3.23 (1.99–5.35)	<0.0001	n.s.	n.s.	2.02 (1.13–3.69)	0.019	n.s.	n.s.
Residual tumor	absent/ present	2.67 (1.34–5.83)	0.008	n.s.	n.s.	1.53 (0.66–3.75)	0.334	n.s.	n.s.
Lymphovascular invasion	absent/ present	11.32 (6.24–21.42)	<0.0001	**6.09** **(3.10–12.30)**	**<0.0001**	7.56 (3.70–16.23)	<0.0001	**5.11** **(2.30–12.02)**	**<0.0001**
Perineural invasion	absent/ present	2.21 (1.19–4.35)	0.016	n.s.	n.s.	3.11 (1.37–7.75)	0.009	n.s.	n.s.

## Data Availability

All data generated or analyzed during this study are included in this published article and its Appendix A or available following reasonable requests from other authors or investigators.

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
