# Peer review of "Prognostic Ability of Tumor Budding Outperforms Poorly Differentiated Clusters in Gastric Cancer"

_cancers, 2022, doi:10.3390/cancers14194731_

Round 1
Reviewer 1 Report
This study was planned to evaluate the Prognostic value of histological phenomena tumor budding (TB) and poorly differentiated clusters (PDCs). I think it is very interesting study, but they need minor revision.
1. It is necessary to clarify the grade of TB and PDC. Each grade should be marked with a clear photo and an accurate arrow to recreate the figure so that readers can recognize it accurately.
2. it is needed to discuss with the previous study (reference [14, 37] of tumor budding and this study.
[14] Gabbert, H.E.; Meier, S.; Gerharz, C.D.; Hommel, G. Tumor-Cell Dissociation at the Invasion Front: A 491 New Prognostic Parameter in Gastric Cancer Patients. Int J Cancer 1992, 50, 202–207, 492 doi:10.1002/ijc.2910500208.
[37] Tanaka, A.; Watanabe, T.; Okuno, K.; Yasutomi, M. Perineural Invasion as a Predictor of Recurrence 563 of Gastric Cancer. Cancer 1994, 73, 550–555, doi:10.1002/1097-0142(19940201)73:3<550::aid-564 cncr2820730309>3.0.co;2-0.
Author Response
„This study was planned to evaluate the Prognostic value of histological phenomena tumor budding (TB) and poorly differentiated clusters (PDCs). I think it is very interesting study, but they need minor revision.”
We are very grateful for the reviewer's critical comments.
„1. It is necessary to clarify the grade of TB and PDC. Each grade should be marked with a clear photo and an accurate arrow to recreate the figure so that readers can recognize it accurately.”
Thank you for this recommendation. Figure 1 has been modified accordingly. This modified figure illustrates the different grades of both tumor budding and poorly differentiated clusters, with arrowheads indicating the individual tumor cell clusters at each grade of tumor budding and PDC.
„2. it is needed to discuss with the previous study (reference [14, 37] of tumor budding and this study.
[14] Gabbert, H.E.; Meier, S.; Gerharz, C.D.; Hommel, G. Tumor-Cell Dissociation at the Invasion Front: A 491 New Prognostic Parameter in Gastric Cancer Patients. Int J Cancer 1992, 50, 202–207, 492 doi:10.1002/ijc.2910500208.
[37] Tanaka, A.; Watanabe, T.; Okuno, K.; Yasutomi, M. Perineural Invasion as a Predictor of Recurrence 563 of Gastric Cancer. Cancer 1994, 73, 550–555, doi:10.1002/1097-0142(19940201)73:3<550::aid-564 cncr2820730309>3.0.co;2-0.”
Thank you for this comment. In accordance with the reviewer's request, the discussion of the scientific message of the two mentioned references has been included in the Discussion section of the manuscript (lines 360-365 and 409-415).
Reviewer 2 Report
This study investigated the association between either tumor budding or poorly differentiated clusters and oncological outcomes among gastric cancer patients. I have some suggestions listed below.
1. In the statistical analysis, I consider if the odds ratio of T staging / N staging relects every-one increment of T or N staging.
2. As to the diagnosis of either tumor budding or poorly differentiated clusters by microscopy, what is the training for the pathologist? I wonder whether most pathologists are familiar with the diagnosis?
3. About the differences between ethnicity, do you consider that the differences of tumor budding or poorly differentiated clusters expressing exist between ethnicity?
Author Response
„This study investigated the association between either tumor budding or poorly differentiated clusters and oncological outcomes among gastric cancer patients. I have some suggestions listed below.”
We very much appreciate the reviewer's helpful comments and questions.
„1. In the statistical analysis, I consider if the odds ratio of T staging / N staging relects every-one increment of T or N staging. „
Thank you for this question. Although our cohort is large compared to similar studies in gastric cancer, if too many subgroups are created along different criteria, the differences between individual subgroups quickly lose statistical power when these criteria are combined. Therefore, as in other previous studies, in order to ensure greater statistical power in the calculation of hazard/odds ratios in multivariable analysis, not all possible increments were included in the model in the case of pT and pN, but to also take into account clinical relevance, pT stages I-II and III-IV were combined, so that a two-stage scale (I-II vs III-IV) was used instead of four, and a similar two-stage (pN0 vs pN+) scale was used for pN (see manuscript references 4, 6 and 35, which studies addressed this issue in the same way). To further clarify that this is how we performed the statistical analysis, we have included a sentence about this in the Statistical analysis part of the Methods section (lines 168-173) and noted it in Tables 3, 4 and 5 as well.
„2. As to the diagnosis of either tumor budding or poorly differentiated clusters by microscopy, what is the training for the pathologist? I wonder whether most pathologists are familiar with the diagnosis?”
As the assessment of tumor budding is part of the routine diagnostic work-up for colorectal carcinomas, all gastrointestinal pathologists are familiar with the evaluation of tumour budding and the criteria set by the ITBCC for counting the tumor buds. The counting of poorly differentiated clusters is identical to the previous method, except for the only difference, which is the number of cells that are present in individual cell clusters. In view of all these considerations, we believe that the introduction of tumor budding and/or PDC into the routine diagnostic work-up of gastric cancer would provide a new histology-based prognostic factor, the evaluation of which is already known from colorectal cancer and can be easily performed by pathologists responsible for the evaluation.
In our study, primary evaluation of cases was performed by pathology residents (L.S. and Á.J.), who were specifically trained by their supervisors to evaluate tumor budding and PDC. In case of substantially discrepant results, the given case was jointly assessed again in a second round involving further experienced investigator(s) (É.K. and G.L.).
„3. About the differences between ethnicity, do you consider that the differences of tumor budding or poorly differentiated clusters expressing exist between ethnicity?”
Thank you for this question. In our retrospective study, it was not possible to take into account the ethnicity of the patients, as Hungarian law prohibits the recording of ethnicity in medical records in order to prevent ethnic discrimination. But considering that the vast majority of Hungary's population belongs to the Caucasian ethnic group, other ethnic groups would not be represented in an evaluable proportion within the patient cohort.
Round 2
Reviewer 2 Report
My comments had been addressed comprehensively. Congratulation to your great work!